# In Vitro Interactions of Amphiphilic Phosphorous Dendrons with Liposomes and Exosomes—Implications for Blood Viscosity Changes

**DOI:** 10.3390/pharmaceutics14081596

**Published:** 2022-07-30

**Authors:** Martina Veliskova, Milan Zvarik, Simon Suty, Juraj Jacko, Patrick Mydla, Katarina Cechova, Daniela Dzubinska, Marcela Morvova, Maksim Ionov, Maria Terehova, Jean-Pierre Majoral, Maria Bryszewska, Iveta Waczulikova

**Affiliations:** 1Department of Nuclear Physics and Biophysics, Faculty of Mathematics, Physics and Informatics, Comenius University, Mlynska Dolina F1, 84248 Bratislava, Slovakia; martina.veliskova@fmph.uniba.sk (M.V.); simon.suty@fmph.uniba.sk (S.S.); juraj.jacko@fmph.uniba.sk (J.J.); patrick.mydla@fmph.uniba.sk (P.M.); cechova60@uniba.sk (K.C.); daniela.dzubinska@fmph.uniba.sk (D.D.); marcela.morvova2@fmph.uniba.sk (M.M.); iveta.waczulikova@fmph.uniba.sk (I.W.); 2Department of General Biophysics, Faculty of Biology and Environmental Protection, University of Lodz, 141/143 Pomorska St., 90-236 Lodz, Poland; maksim.ionov@biol.uni.lodz.pl (M.I.); maria.bryszewska@biol.uni.lodz.pl (M.B.); 3Institute of Biophysics and Cell Engineering, National Academy of Sciences of Belarus (IBCENASB), Akademicheskaya 27, 220072 Minsk, Belarus; maryterekhova@tut.by; 4Laboratoire de Chimie de Coordination du CNRS, 205 Route de Narbonne, CEDEX 4, 31077 Toulouse, France; jean-pierre.majoral@lcc-toulouse.fr; 5Université Toulouse, 118 Route de Narbonne, CEDEX 4, 31077 Toulouse, France

**Keywords:** amphiphilic dendrons, exosomes, liposomes, self-aggregation, lipid bilayer stability, blood viscosity

## Abstract

Drug delivery by dendron-based nanoparticles is widely studied due to their ability to encapsulate or bind different ligands. For medical purposes, it is necessary (even if not sufficient) for these nanostructures to be compatible with blood. We studied the interaction of amphiphilic dendrons with blood samples from healthy volunteers using standard laboratory methods and rheological measurements. We did not observe clinically relevant abnormalities, but we found a concentration-dependent increase in whole blood viscosity, higher in males, presumably due to the formation of aggregates. To characterize the nature of the interactions among blood components and dendrons, we performed experiments on the liposomes and exosomes as models of biological membranes. Based on results obtained using diverse biophysical methods, we conclude that the interactions were of electrostatic nature. Overall, we have confirmed a concentration-dependent effect of dendrons on membrane systems, while the effect of generation was ambiguous. At higher dendron concentrations, the structure of membranes became disturbed, and membranes were prone to forming bigger aggregates, as visualized by SEM. This might have implications for blood flow disturbances when used in vivo. We propose to introduce blood viscosity measurements in early stages of investigation as they can help to optimize drug-like properties of potential drug carriers.

## 1. Introduction

Dendrimers are branched polymeric globular structures composed of synthetic molecules, dendrons. They are generally accepted as a fourth class of polymers [1,2]. Amphiphilic dendrons consist of a polar/hydrophilic head and a non-polar/hydrophobic tail [3]. Their size depends on the number of repetition cycles, called generations. Dendrons and dendrimers have raised interest for their potentially efficient way of delivering drugs to cells [1,4]. Drug molecules can be trapped in inner cavities of polymers or chemically conjugated to their surface [1,2,3,4,5]. Furthermore, ligands can be bonded to nanoparticles or intercalated into the structure by covalent linkage with the polymeric backbone [5]. The amphiphilic phosphorus dendrons used in our study are a unique combination of hydrophobic pyrene groups modified by hydrophilic branches. The pyrene groups facilitate the interaction of hydrophobic drugs (for example, 5-fluorouracil) and their localization inside dendrons. Thus, these dendrons are promising for efficient delivery of hydrophobic drugs to target cells. On the other hand, cationic surface groups enhance the solubility of dendrons. In addition, the positively charged surface can be used for delivering anionic siRNA or microRNA inside cells, after the non-covalently bound dendrons are self-destroyed in endosomes or lysosomes and eliminated from a cell [1,5]. However, in order to develop dendrons for use in drug delivery to cells, a better understanding of how these macromolecules interact with the biological systems is required. In particular, blood compatibility of the compounds must be ensured.

Whole blood viscosity is known to be a strong function of the number (and volume) of formed elements, especially red blood cells (RBCs), and is thus positively related to hematocrit [6,7,8]. Since about 45% of blood volume is made up of suspended cellular particles, primarily red blood cells, the blood behaves as a non-Newtonian fluid whose viscosity varies with shear rates (i.e., with the rate of increase in blood flow velocity of adjacent streaming layers) [9]. When blood moves at a high velocity, blood viscosity becomes relatively low, since RBCs are maximally deformed and dispersed to reduce flow resistance. This characteristic of blood is known as shear thinning behavior. On the other hand, RBCs have a propensity to aggregate and stack together when they are at rest or when they flow at low shear rates, which is caused by the interactions between RBCs and molecules present in blood plasma, mainly fibrinogen [10]. The attraction is attributed to charged groups on the surface of the cells and molecules. The viscosity of blood thus depends highly on the RBC concentration and biomechanical properties, such as aggregation–deaggregation properties and membrane elasticity [6,7,11]. Blood viscosity increases exponentially as RBCs tend to aggregate. Therefore, processes which affect the interplay of aggregation-deaggregation of RBCs, including increases in the levels of RBC and platelet aggregation from any cause, can disturb blood flow with potentially serious health consequences [12]. Indeed, experimental and clinical data have clearly shown that the flow behavior of blood is a major determinant or risk factor of cardiovascular diseases linked to atherosclerosis (such as coronary artery disease, cerebrovascular disease) and to microvascular disorders, including diabetic retinopathy, nephrophathy, and neuropathy [13,14,15]. Increased blood viscosity has also been linked to several hereditary diseases such as sickle cell anemia [16]. Amphiphilic dendrons, if used as drug carriers, must not considerably disrupt blood flow; therefore, it is important to gather information about their aggregation propensity.

Controlled experiments on model membranes can bring valuable information about the nature of interactions among membrane blood components and dendrons. Liposomes are one of the possible synthetic model membrane structures. These spherical nanoparticles are composed of amphiphilic molecules, predominantly phospholipids, forming a phospholipid bilayer. Due to their amphiphilic nature, they are able to self-assemble into spherical particles, and thus encapsulate hydrophilic drug molecules dissolved in the liquid [5]. Liposomes have potential in drug delivery for many reasons, including their similarity with cell membrane (structure and composition). The great advantage is their biocompatibility, biodegradability, as well as easy preparation and stability over time [5,17,18,19]. According to Tomar et al. [5], liposomes tend to be sterically unstable and thus are cleared rapidly from the bloodstream. On the other hand, liposomes can be coated with polymeric coating, which enhances circulation time and bioavailability of the encapsulated drug, thus increasing target efficiency [5]. Liposomal nanoparticles easily diffuse across the cell membrane and can release the encapsulated drug within cells [5]. There are various methods used to prepare liposomes, such as sonication, evaporation, solvent injection, reverse phase, and extrusion [5,20,21].

A fundamental physicochemical property of liposomes is membrane fluidity, which describes the dynamics, microviscosity, and order of lipids in the lipid bilayer (liposomes, exosomes). It can influence the final therapeutic efficiency of the loaded drug [22]. Membrane fluidity can be altered by using phospholipids with different phase transition temperature T_c_, which is the temperature where arrangement of lipids changes from well-ordered (solid gel-like phase) to liquid crystalline (fluid phase) and vice versa [23]. The phase transition temperature depends on the number of saturated or unsaturated bonds in hydrophobic acyl chains of phospholipids, or its length [24,25,26]. Increased length of fatty acid tail or increased amount of saturated fatty acids decreases the fluidity of the membrane [26]. Generally, with temperature above T_c_, the fluidity of the membrane increases, leading to a more permeable membrane [27]. The presence of high T_c_ lipids (T_c_ > 37 °C) makes liposome bilayer membrane less fluid (i.e., long-chain lipids, membranes rich in saturated fatty acids [26]) at physiological temperature and therefore less leaky. On the other hand, liposomes composed of low T_c_ lipids (T_c_ < 37 °C, i.e., short-chain lipids, membranes rich in unsaturated fatty acids [26]) are more susceptible to leakage of drugs encapsulated in the aqueous phase at physiological temperature [23]. When interacting with cells, liposomes containing high T_c_ lipids appear to have a lower extent of uptake by reticuloendothelial system compared to liposomes with low T_c_ lipids [23]. Fluidity is also dependent on the cholesterol concentration in the membrane [23,26,27].

Other model membrane structures are exosomes, which are naturally present in all body fluids, thus representing a natural membrane system composed of lipids and other biologically active macromolecules, such as proteins. These extracellular vesicles occur naturally in the body and their size ranges from 30 to 150 nm, similar to liposomes. Exosome generation starts with forming an endocytic vesicle originating from inward budding of plasma membrane, which is sorted into an early endosome [28]. Early endosomes mature into late endosomes, also known as multivesicular bodies (MLVs). Within MLVs, intraluminal vesicles (ILVs) are formed [29]. ILVs are generated by the inward budding of endosomal membranes [28,30]. Most ILVs are released into the extracellular space, which are referred to as exosomes [30]. The membrane of exosomes consists of a lipid bilayer, but the composition differs from the cell membrane in the relative proportion of individual lipids. Exosomes are enriched in sphingomyelin, gangliosides, and unsaturated lipids, and the proportion of phosphatidylcholine and diacylglycerol is lower than in the cell they originated from. Studies also show an increased proportion of cholesterol in exosomes compared to cell membranes, as well as an increased proportion of phosphatidylserine [31,32,33]. The main difference between liposomes and exosomes lies in the proteins (mainly tetraspanins) that are present in the membrane of the exosomes. These proteins are important for targeted uptake by recipient cells. Exosomes have also unique lipid composition, different from lipid composition of liposomes, which may be important in the incorporation and functional conformation of the proteins in their membrane [34].

Charged nanoparticles based on polymers, such as dendrimers, and the study of their interactions with model membranes provide important information about the effect of foreign molecules on liposome structures [35]. Despite the significant increase in their biological applications documented in the past decade, some problems remain unresolved, which are mainly related to the possible disruptive and degrading effects of interactions with biological membranes, which may be associated with undesirable cytotoxic effects and low biocompatibility [36]. In an in vitro study, we aimed to assess the potential hemorheological impact of the amphiphilic phosphorous dendrons on whole blood in young healthy subjects. In parallel, we have explored the interactions of dendrons with model membrane systems in the controlled experimental conditions using liposomes and exosomes. To describe the interactions, we have employed various physical methods, i.e., high-performance liquid chromatography, dynamic light scattering, fluorescence anisotropy, and scanning electron microscopy.

## 2. Materials and Methods

### 2.1. Chemicals and Dendrons

We used 10 mM phosphate buffer, pH 7.4, adjusted with 1 M HCl or 1 M NaOH at 25 °C. 1,2 dimyristoyl-sn-glycero-3-phosphocholine (DMPC), 1,2-dimyristoyl-sn-glycero-3-phospho-rac-(1-glycerol), sodium salt (DMPG), and cholesterol (Chol) lipids were purchased from Sigma-Aldrich (Darmstadt, Germany). In this work, amphiphilic phosphorus dendrons of the first (D1) and second (D2) generation were prepared and studied according to [37] (Figure 1). The chemical structure and molar mass of the first-generation dendron were C_138_H_224_C_l10_N_34_O_7_P_8_S_5_ and Mw = 3234.13 g/mol, and the chemical structure of the second generation was C_268_H_424_C_l20_N_74_O_17_P_18_S_15_ with a molar mass Mw = 6702.27 g/mol. Phosphate buffer, pH 7.4, purchased from Sigma-Aldrich (Darmstadt, Germany), was used to isolate the exosomes.

### 2.2. Preparation of Dendrons

A stock solution of dendrons dissolved in Milli-Q water at a concentration of 3 mM was prepared. From this stock solution, dendrons were added to the liposomes so that their final concentration was 1 µM, 5 µM, 10 µM, 15 µM, 20 µM and 50 µM when measuring the concentration dependence. When carrying out time measurements, the concentration of dendrons in 15 µM in liposomes and the concentration of dendrons in exosomes was 2 µM.

### 2.3. Preparation of Liposomes

For liposome synthesis, a mixture of DMPG, DMPC, and cholesterol (DMPG-DMPC-Chol) in the molar ratio 63:7:30 was used to assemble the liposomes. DMPG represented 10 mol % of DMPC, and Chol was added in an amount of 30 mol % of the DMPC and DMPG mixture. The dry lipid film was hydrated with 10 mM Na-phosphate buffer, pH 7.4, to prepare a lipid solution at a concentration of 10 mg/mL. Liposomes were extruded through a polycarbonate membrane using a heating block mini extruder (Avanti Polar Lipids Inc., Alabaster, AL, USA) with a pore size of 400 nm to obtain unilamellar liposomes.

### 2.4. Exosomes Isolation

Exosomes were isolated from urine as well as from blood. For isolation, we used the ultrafiltration method in combination with size exclusion chromatography.

Isolation of exosomes from urine involved three steps of centrifugation at gradually increasing relative centrifuge force (200 g, 2000 g, 16,000 g) to remove cells, bacteria, and proteins. We separated the supernatant from sediment, and then added 1.3 M DL-dithiothreitol (DTT) (Sigma-Aldrich, Germany) to the sediment to remove protein uromodulin, which is normally physiologically present in urine. After a subsequent 10 min incubation, 10 mM solution of TRIS (Sigma-Aldrich, Germany) with 250 mM sucrose (Sigma-Aldrich, Germany), pH 7.6, was added to the suspension, and the suspension was further centrifuged at 16 680 g. After this centrifugation, the supernatant was combined with the supernatant, which was separated after centrifugation at 16 000 g, and those mixed supernatants were filtered through a 0.22 μm pore size filter. Isolation of exosomes from blood also involved three steps of centrifugation at gradually increasing relative centrifuge force (700 g, 1500 g, 12,000 g) to remove blood cells, platelets, and microvesicles. After the centrifugations, supernatants were filtered through a 0.22 μm pore size filter.

From this step, the isolation process was the same for both blood and urine samples. The filtered supernatants were concentrated to the required volume of retanate (ca. 100 µL), and molecules smaller than 300 kDa were removed by nano-filters (VivaSpin, 300 kDa). Subsequently, the retanate prepared in this way was injected into the HPLC system. In the HPLC system, we used an SRT SEC-300 column with dimensions of 7.8 × 300 mm and particle size of 5 μm, which would allow the separation of particles present in the retanate based on size. Exosomes, as the largest particles, were eliminated from the column first, at a flow rate of 0.4 mL/min from 12 to 20 min. The collected fractions were then concentrated through a 30 kDa VivaSpin filter.

In our study, exosomes isolated from blood and urine were used as a natural model of cell membranes. Because using blood as a source of exosomes was not always possible due to large volume of samples required, especially for chromatographic measurements, we used urine that is an easily available body fluid, even in large volumes. Moreover, exosomes isolated from urine are also more homogeneous than those from blood (exosomes transported in blood have their origin in various organs; thus, the resulting yield would be more heterogeneous). However, for scanning electron microscopy, where only small volumes of sample are required, exosomes isolated from blood were used to make visualization experiments more consistent with blood compatibility studies.

### 2.5. The Blood Drawing

Fasting blood specimens were collected from 12 volunteers aged 20–30 years (6 males and 6 females) in good health condition without serious health issues. The blood drawings were performed by a nurse at the Department of Hematology and Transfusiology of the St. Elizabeth Cancer Institute in Bratislava. Blood was drawn in the morning hours from the elbow vein into one 10 mL tube with the anticoagulant K2EDTA with the final concentration of 1.8 mg/mL (BD Vacutainer^®^, ref. n. 367525). Standard hematology and coagulation tests for all samples (in the presence and absence of D1 or D2) were performed in accordance with the standard methods directly in the laboratory. No clinically relevant abnormalities were detected, nor was any apparent isotonic hemolysis observed immediately after the addition of D1 and D2 dendrons. This study was conducted in accordance with the ethical principles set forth in the Declaration of Helsinki and the study protocol was approved by the Institutional Ethics Committee at the St. Elizabeth Cancer Institute in Bratislava (Approval No. 03-2020/EK OÚSA, signed on 4 March 2020).

### 2.6. Viscosity Measurements

Immediately after blood drawing, measurements of rheological properties of blood were performed at the Institute of Medical Physics, Biophysics, Informatics and Telemedicine, Faculty of Medicine, Comenius University in Bratislava on a modular compact rheometer MCR 102 with double gap measuring system DG 26.7/ Ti (Anton Paar, Graz, Austria).

The tubes of blood were placed in a water bath with thermostat Polystat Control cc1 (Huber, Berching, Germany) at constant temperature of 37 °C and the tubes were manually agitated by slowly inverting them several times every 10 min. Subsequently, 4990 µL of blood was pipetted into the cup of the double gap measuring system. After the first measurement of rheological properties of pure blood, we titrated 10 µL from the dendron D1 solution of c_D1_ = 1 mM, giving a final concentration of c_1_ = 2 µM. After the measurements were made, another 40.08 µL of the 1 mM D1 solution was titrated into the sample to achieve a final concentration of c_2_ = 10 µM. We followed the same procedure using dendron D2, with the blood measurement without added nanoparticles.

All measurements were performed at a temperature of 37 °C, maintained by the Peltier system built into the MCR 102 rheometer itself, at the three shear rates—100 s^−1^, 10 s^−1^, and 1 s^−1^. The length of the measurement for a given point was determined automatically by the instrument until a stable viscosity value was reached at that point.

### 2.7. Chromatographic Measurements

Chromatographic measurements were performed using high-performance liquid chromatography (HPLC) Prominence 20A system (Shimadzu co., Kyoto, Japan), which, in addition to component separation, allowed us to measure the absorption signal from liposomes, urinary exosomes, and dendrons. SRT SEC-300 chromatographic column (particle size 5 µm, pore size 300 Å, 7.8 × 300 mm) (Sepax Technologies, Newark, DE, USA) was used to separate individual components of the sample, and 10 mM phosphate buffer was used as a mobile phase, pH 7.4, at a flow rate of 0.5 mL/min. Absorption detector wavelengths were set to 240 nm. Cholesterol present in both liposomes and exosomes has an absorption maximum at 240 nm; thus, the signal (absorbance) represents the amount of nanoparticles (containing cholesterol) at the given time in the detector cell. Temperature in the thermostat was maintained at 37 °C, and the volume of the injected sample was 100 µL.

To monitor the interaction of liposomes (c = 1 mg/mL) and urinary exosomes, size exclusion chromatography (SEC) with absorption detection was used. Dendrons were added to membrane structure samples at concentration of 2 µM for both generations of dendrons (D1 and D2). Spectral properties were measured immediately after adding dendrons to the liposomes or urinary exosomes and subsequent incubation at 37 °C for one, two, and four hours. Size exclusion chromatography (SEC) separates molecules/particles based on their size. Larger molecules/particles elute from the column sooner (do not enter the pores) and smaller molecules/particles elute later (penetrate deep into the pores), which effectively sorts the molecules/particles by size.

### 2.8. Fluorescence Anisotropy Measurements

To investigate the impact of interaction between dendrons with phospholipid bilayer, the measurement of fluorescence anisotropy using the fluorescence probe 1,6-diphenyl-1,3,5-hexatriene (DPH, Sigma-Aldrich, St. Louis, MO, USA) was performed. Measurement of the steady-state anisotropy of DPH probe is the most common technique for evaluation of ordering of the membrane lipids. The DPH stock solution was dissolved in acetone and stored in the fridge at 4 °C. Before each measurement, the staining solution was prepared, dissolved in PB (pH 7.4), and shaken for 20 min to evaporate acetone. The final concentration of DPH in suspension was 1.5 × 10^−7^ M. A luminescence spectrometer LS 45 (PerkinElmer, Waltham, MA, USA) was used for L-format polarization measurements of the steady-state fluorescence anisotropy of DPH. The excitation and emission wavelengths were set to 360 and 430 nm. The measurements were carried out at 37 ± 0.1 °C. Liposomes were incubated with dendrons for 15 min before each measurement. Grating correction factor (*G*) for the optical system was measured, which is the ratio of the sensitivities of the detection system for vertically and horizontally polarized light [38], where:(1)$$G=\frac{I_{HV}}{I_{HH}}$$

*I_HV_* and *I_HH_* represent the fluorescence intensity when the excitation polarizer is in the horizontal position and when the emission polarizer is in the vertical or horizontal position. Every measurement lasted for 2400 s. The steady-state fluorescence anisotropy (*r_s_*) was used for the assessment of membrane fluidity and calculated according to the following equation [38]:(2)$$r_{s}=\frac{I_{VV}-GI_{VH}}{I_{VV}+2GI_{VH}}$$

Lower values of fluorescent anisotropy represent higher membrane fluidity.

### 2.9. Size Measurements

The size of the prepared liposomes upon the interactions with D1 and D2 dendrons was measured by a dynamic light scattering technique (DLS) using a Zetasizer-Nano ZS90 spectrophotometer (Malvern Instrument, Malvern, UK). The concentration of lipid vesicles was 1 mg/mL. A stock solution of dendrons dissolved in Milli-Q water at a concentration of 2 mM was prepared. Dendrons were titrated into the liposomal solution by adding small aliquots from stock solution or one of two working solutions prepared from the stock solution. The first working solution was a 0.05 mM solution used for a final dendron concentration of 0.1 μM; the second was a 0.5 mM solution used for preparing samples with final concentration of 1 μM. Stock solution was used as working solution for preparing higher final dendron concentrations (3 μM, and 10 μM). The hydrodynamic size parameter, known as Z-average size, was analyzed using Malvern software. All measurements were performed at 37 °C.

### 2.10. Scanning Electron Microscopy

D1 and D2 dendrons were added to samples of liposomes and exosomes isolated from blood according to [39,40,41] in concentrations of 2 and 10 µL. Incubation time was 1 h. Exosomes and liposomes were then fixed with 3.7% glutaraldehyde (Electron Microscopy Sciences, Hatfield, PA, USA) in phosphate buffer (PBS, Sigma-Aldrich, Germany) for 15 min. After washing twice with PBS, the samples were dehydrated with ascending sequence of ethanol (40%, 60%, 80%, 96%). Afterwards, a drop of the sample (1 µL) was applied to a double-sided adhesive carbon tape applied to a stainless-steel pin (∅ 9.5 mm) and the samples were left to dry at room temperature for 30 min. Analysis by scanning electron microscopy (SEM, ZEISS EVO LS15) followed and was performed at 3 different points in each sample [42]. Exosomes were captured in a secondary electron (SE) imaging mode and liposomes in a backscattered electron (BSD) imaging mode. Different imaging modes and inverting of micrographs were chosen for the better contrast.

### 2.11. Statistical and Graphical Analysis of Results

To model the shear rate dependent viscosity, we evaluated performance of several constitutive models, and finally selected the Quemada model to fit the data. To examine whether the new class of amphiphilic phosphorous dendrons affect blood viscosity, we used an experimental setup with three kinds of independent variables (factors), each with two levels: generation of dendrons (D1 and D2), concentration of dendrons (2 μM and 10 μM), and subject’s sex (male and female). We hypothesized that the dendrons might affect males and females differently; therefore, a three-way between-subjects ANOVA (analysis of variance) was used to analyze blood viscosity data [43]. In variable terms, we were interested in whether there was some sort of interaction between generation, concentration, and sex on blood viscosity. Interaction plots were used to visualize the outcomes and evaluate the pattern of relationship.

We used the LCsolution program, version 1.25, to analyze the chromatographic results. We exported measured data to MS Excel, where we made chromatograms. Subsequently, we used the R program to create contour plots for a better visual comparison of the results. The Kruskal–Wallis chi-squared test was used to compare size of liposomes at various dendron concentrations. To perform the statistical analysis, GraphPad Prism 9.3.1 (GraphPad Software, San Diego, CA, USA) was used.

## 3. Results

Firstly, we evaluated the interaction of dendrons (D1 and D2) with membrane structures in terms of change in their size by chromatographic separation, absorption detection, and the DLS method. We monitored the change in size of membrane structures depending on the added concentration of dendrons, as well as the length of incubation time. Secondly, we measured changes in blood viscosity after adding three different concentrations of two generations of dendrons. Finally, intensities of fluorescence anisotropy were obtained to investigate the impact of interaction between dendrons and liposomes as model membrane structures. Images of exosomes after the addition of dendrons were obtained by scanning electron microscopy. All measurements were carried out at 37 °C.

### 3.1. Chromatographic Analysis of Membrane-Dendron Interaction

Figure 2a shows the absorption chromatograms of dendron D1 with liposomes in terms of concentration effect. A concentration of 0 µM indicates liposome alone without added dendrons, which has its main peak at a retention time (RT) of 11 min. Peaks after the addition of dendrons have a maximum intensity in RT of 10 min. We see that the signal intensity increases to a concentration of 10 µM, and then the signal begins to decrease, while moving to the left. This is also confirmed by the nested graph, which plots the absorbance ratio of a given peak at 11 and 10 min.

Figure 2b shows contour plots with liposome absorbance values at different concentrations of D1 and D2. In the second generation of dendrons (bottom), with increasing concentration, the absorbance decreases and becomes undetectable by our detector compared to the first-generation dendron (top) at the same retention time. Moreover, in the second generation of dendrons (D2), the signal is shifted to the left (larger particles are eluted earlier).

Figure 3a shows the absorption chromatograms of dendron D1 at a concentration of 15 µM with liposomes after incubation at 37 °C in terms of the time effect. As the incubation time increases, the absorbance signal and the area under the peak also increase, which is shown in the nested graph. In addition, the peak shifts to the left (larger particles are eluted earlier).

Figure 3b again compares the first- and the second-generation dendrons with different incubation times with liposomes using contour plots. The absorbance signal can be observed during the four hours of incubation for D1, while for D2, we did not detect the signal immediately after the addition of dendrons to the liposomes, nor in the following hours.

Figure 4a shows the shift and decrease in absorbance intensity after adding dendron D1 at a concentration of 2 µM to exosomes. The signal disappears at RT 22 min. The absorbance ratio at 22 and 11 min decreases with incubation time.

Figure 4b compares the contour plots of D1 and D2 dendrons after incubation with exosomes isolated from urine at 37 °C at different time points. The difference between the graphs is best seen at RT 11 min, where we observe a weak signal for dendron D1, which we did not detect for dendron D2. Furthermore, even at RT 22 min, we observe a weaker signal with dendron D2 over time.

### 3.2. Size Measurements

In addition to separation of liposomes and exosomes with dendrons by size exclusion chromatography, we also measured the size of pure liposomes and liposomes after the addition of D1 and D2 by the dynamic light scattering technique (DLS). Three independent replicates were obtained for each experimental situation. Boxplots shown in Figure 5 represent the size of liposomes with D1 and D2 dendrons at concentrations of 0 µM, 3 µM, and 10 µM. Liposomes without dendrons had an average size of 360 nm. The addition of dendron D1 led to an increase in size proportional to the concentration of dendrons (Z-size average 388 nm and 479 nm for c = 3 µM and 10 µM, respectively). The increase in size was even more obvious after adding second-generation dendron D2. The Z-average size was 527 nm and 2860 nm for added concentration of dendrons of 3 µM and 10 µM, respectively. The Kruskal–Wallis chi-squared test was used in both cases with *p* < 0.001.

### 3.3. Viscosity Measurements

Blood compatibility is necessary for effective and safe drug delivery in patients. Therefore, in parallel to the measurements on model membranes, we carried out in vitro measurements of whole blood viscosity depending on dendron generation, sex, and concentration for three different shear rates, 1 s^−1^, 10 s^−1^, and 100 s^−1^, selected to simulate flow conditions at low shear rates (as in veins) through middle to high rates (as in arteries) (Figure 6). Concentrations of added D1 and D2 dendrons were 0 µM, 2 µM, and 10 µM.

We confirmed a linear relationship between hematocrit and hemoglobin content in RBCs. Subject’s sex did not interact statistically with hemoglobin to predict hematocrit (the ANCOVA test of slopes was performed). Thus, the stratification by sex sufficiently addressed dependence of whole blood viscosity on hematocrit in the analysis.

An increase in viscosity was positively associated with the increase in the concentration irrespective of dendron generation at all selected shear rates: 1 s^−1^ (*p* = 0.0002), 10 s^−1^ (*p* = 0.0001), and 100 s^−1^ (*p* = 0.0003). In Figure 6, we can see the shear-thinning behavior of blood—the viscosity is higher for lower shear rates. It is also seen that, for a particular shear rate, the viscosity is higher for males. The observed difference was statistically significant (*p* = 0.001, *p* = 0.002, *p* = 0.014, respectively). This result also means that the impact of hematocrit on blood viscosity is much higher at low shear rates than at high shear rate. By this analysis, the main effect of the factor sex outweighed that of concentration within the tested concentration range. No significant interaction was identified between the factors sex and concentration, which means that the factors were additive in their effects. The third main factor, dendron generation D1 or D2, was not significant and did not contribute to the viscosity alterations (for more details, see Appendix A).

### 3.4. Fluorescence Anisotropy

Fluidity of membrane is another important biophysical property of membranes which is affected by the presence of dendrons. We observed changes in anisotropy of fluorescence, which is inversely proportional to fluidity. The concentrations of added dendrons were 0 µM, 2 µM, and 10 µM, as shown in Table 1. Fluorescence anisotropy was affected by D1 and D2 in a similar manner. However, a higher concentration of dendrons had visually greater effect on liposomes than a lower concentration.

### 3.5. Scanning Electron Microscopy

Using SEM, we captured pictures of liposomes and exosomes isolated from blood according to [39,40,41] (control) and liposomes and exosomes with D1 and D2 dendrons at concentrations of 2 and 10 µM. For exosomes, images were magnified 50,000× (Figure 7). Aggregation of exosomes and liposomes was visible after adding dendrons in a dose-related manner.

## 4. Discussion

The study of dendrons and their interactions with membrane structures is an important part of the research on delivering drugs into cells using nanoparticles. In vitro experiments are necessary to verify the biosafety of nanoparticles, and observing their effect on cell membrane models allows us to move closer to developing a safe and effective carrier.

To investigate the effect of first (D1) and second (D2) generation dendrons on membrane-containing biological systems, we used liposomes and exosomes isolated from urine as models. We monitored a change in the absorption signal depending on the change in the concentration of dendrons in the sample (Figure 2 and Figure 4), as well as in terms of time by measuring the samples immediately after the addition of dendrons, and subsequently after one, two, and four hours of incubation at 37 °C (Figure 3). After interaction of D1 dendrons with liposomes, we observed an increase in absorbance with increasing concentration up to 10 µM. On the contrary, at higher concentrations, the absorbance decreased (Figure 2). In the D2 generation, we observed a similar but stronger effect, as in this generation of dendrons, the absorbance decreased already at a concentration of 1 µM, and at a concentration of 5 µM, the absorption signal was minimal. However, the maximum signal was significantly shifted to the left towards the earlier retention time (Figure 2b). Aggregation of membrane structures—liposomes as well as exosomes—isolated from blood was also observed by scanning electron microscopy (Figure 7). It appears that the size of aggregates was not different between generations; however, there were differences associated with concentration of dendrons in the preparation.

Most studies have focused on the interactions between positively charged dendrons or dendrimers and cell membranes because greater interaction potency is expected in comparison with other neutral or negatively charged nanostructures. Physico-chemical properties of an amphiphilic dendron influencing the dendron–lipid bilayer interaction include the type of the dendron, generation, surface charge, the composition of the membrane, and the solvent. We have already proved these relationships, including that the interaction of dendron-based nanoparticles with liposomes is of electrostatic nature, in our previous works (e.g., [1,44]). It can be concluded that neutral DMPC liposomes do not interact with dendrons; on the other hand, large aggregates are observed when interacting with negatively charged DMPG-DMPC-Chol liposomes [1,37]. According to Janaszewska et al. [45], cytotoxicity depends directly on the concentration and generation of dendrons. The interaction between positively charged nanoparticles and a negatively charged membrane leads to the formation of nanopores in the membrane and its damage [45]. In addition, as the concentration of dendrons in the sample increases, we observed a shift in the chromatograms to the left (since larger particles are eluted earlier), so we assume that a spontaneous aggregation of the disrupted membranes and formation of larger structures might have occurred. Based on the work of Janaszewska et al. [45], we assume that the effect of generation of dendrons on the liposome size could be explained in a similar manner. Thus, the oppositely charged dendrons act as glue to promote aggregation. Aggregation of liposomes was also supported by DLS measurements, where the size grew proportionally to both dendron generation and concentration (*p* < 0.001) (Figure 5), and the prevalence of electrostatic interactions was evaluated through the zeta potential measurements.

Investigation of the effect of time on the absorption properties and size of dendrons with liposomes was performed at 15 µM, representing the upper limit of the concentration range used in the cell studies (Figure 3). For urinary exosomes that more closely mimic the natural membrane pattern, we decided to use the same concentrations of 2 µM and 10 µM for both D1 and D2 dendrons, as previously reported (Figure 4) [1]. The effect of dendrons on liposomes was greater with increasing incubation time, especially in terms of the particle size (Figure 3). Over time, the absorbance increased and shifted to the left. A shift to the left points to a possible enlargement of the molecule, as larger particles are eluted from the column earlier. This is most visible on the absorption chromatogram of dendron D1 with liposomes. With D2 dendrons, the effect was similar to that with concentration dependence—the signal practically disappeared immediately after the addition of dendrons.

Measurement of the signal from exosomes with D1 dendrons at the concentration of 2 µM indicates disruption of the urinary exosomes, as the signal was decreasing with time over 22 min (Figure 4). According to the contour graphs, in addition to the disruption of exosomes, it would also be possible to consider their aggregation or association with dendrons and the formation of larger complexes based on the observed signal in 11 min (Figure 4). The explanation for the underlying mechanism of the interaction of exosomes with dendron-based nanoparticles comes with many uncertainties, because of insufficient knowledge of the nature of exosomes and their role in the (patho)physiology of overall health [46]. However, an understanding of exosome interaction with dendron-based nanoparticles can provide novel insights into a new, mostly unexplored area of exosome-mediated delivery of currently used drugs or other promising therapeutic agents.

In the past couple of decades, much attention has been focused on abnormal (increased) whole blood viscosity and its contribution to the pathophysiology of coronary heart disease and microvascular disorders [13,14,15]. It is therefore important to garner information about aggregation propensity of amphiphilic dendrons and other compounds widely studied for medical applications. Such information will greatly help to engineer their properties for selective and safe delivery of drugs, nucleic acids, and other relevant agents [3,47,48]. It is a well-established fact that blood is a non-Newtonian fluid whose viscosity depends on several parameters: number, size, morphology, and mechanical properties of RBCs (membrane fluidity and elasticity); plasma chemistry and protein content; and interactions among whole blood components (aggregation–disaggregation properties) [6,7,49]. At low shear rates (less than 1 s^−1^), RBC aggregation is less likely to occur due to the higher rigidity of RBCs, leading to a decrease in blood viscosity. On the other hand, at shear rates above 1 s^−1^, which are common in microcirculation [50], a decrease in RBC deformability leads to an increase in blood viscosity [16]. Dendrons also affect membrane integrity that is accompanied by leakage of hemoglobin [51]. Mean cell hemoglobin concentration affects internal (cytosolic) viscosity, which has an impact on RBC deformability and blood viscosity [16]. Moreover, it is well known that men have generally higher blood viscosity than women due to their higher volume percentage of red blood cells in blood (hematocrit). The impact of hematocrit on blood viscosity is much higher at low shear rates than at high shear rates [12,16]. After adding dendrons in blood samples, we observed increasing viscosity at different shear rates (1 s^−1^, 10 s^−1^, 100 s^−1^), but the difference was not statistically significant between the two generations of dendrons (Figure 6). This increase in viscosity implies that the formation of aggregates is proportional to concentration of used dendrons, which is in accordance with our previous findings on other amphiphilic dendrons [1].

Since blood viscosity depends not only on the concentration and aggregation of RBCs, but also on the elasticity of their membranes [6,12,50], membrane fluidity estimation can suitably supplement the rheological measurements and, at the same time, bring additional information on the character of interaction between the tested compound and membrane systems. Membrane fluidity is dependent on many factors, such as type and order of lipids in the membrane or membrane composition as a whole [22]. This physicochemical property plays an important role in drug delivery. We tested the effect of dendron generation and concentration on liposomal membranes. Similarly, as in viscosity measurements, we did not observe any difference between dendron generations in terms of their effect on membrane fluidity. This can be, at least partly, explained by similarities in the chemical structure of D1 and D2, which both have the same non-polar parts of molecules, while polar parts differ [42]. We presume that the non-polar parts of planar dendron molecules (their tails) incorporate in the membrane, thus disturbing membrane integrity. Polar parts (hydrophilic heads) interact with water; they “stick out” of the membrane and do not affect the membrane bilayer significantly. On the contrary to the effect of generation, membrane fluidity was affected proportionally to dendron concentration, which is consistent with the previous findings. A plausible explanation is that with high concentration, more dendrons incorporate in the membrane bilayer and disturb lipid packing.

Overall, it can be argued that in addition to standard laboratory tests, investigation of the effects of amphiphilic dendrons (or any engineered dendron-based structures) on membrane fluidity, deformability, and rheological behavior of blood can deliver valuable information on the interactions between the formed blood elements and the applied compound. Such early information can help assess which structures and conditions are most likely disturb blood flow, and which are safe in terms of hemorheology. Analyzing different scenarios may contribute to optimizing drug-like properties of a compound series.

## 5. Conclusions

Understanding molecular interactions is fundamental to improve the biological activity of dendron-based polymers tested for medical use. In our study, we focused on the interaction of different concentrations and generations of amphiphilic phosphorous dendrons with blood and model membranes. Using various physical techniques, we observed aggregation of membrane structures, most likely due to the electrostatic nature of the interaction between dendrons and membranes. This aggregation was concentration-dependent, while the effect of dendron generation was not apparent. Changes in blood viscosity associated with dendron generation and concentration follow the same pattern; however, the increase induced by dendrons at low blood velocities did not reach baseline viscosity values of males who naturally have a higher number of red blood cells. Therefore, this effect is without physiological relevance, which points to good hemocompatibility of amphiphilic dendrons within the tested concentration range. Experiments on model membranes confirmed that higher concentrations of dendrons begin to exert a disturbing influence and can negatively affect membrane integrity. Our study has introduced a novel approach to investigating hemocompatibility of tested dendron-based polymers on whole blood, which can be used for better understanding how these macromolecules interact with biological systems, and which can consequently help further optimize their structure and function in vivo.

## Figures and Tables

**Figure 1 pharmaceutics-14-01596-f001:**
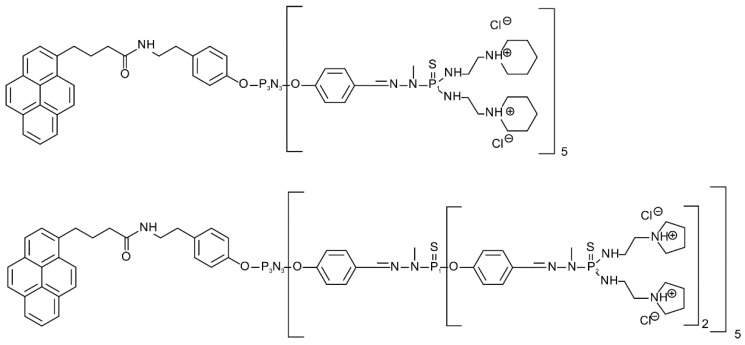
Structure of first-generation dendron, D1 (**top**); second-generation dendron, D2 (**bottom**).

**Figure 2 pharmaceutics-14-01596-f002:**
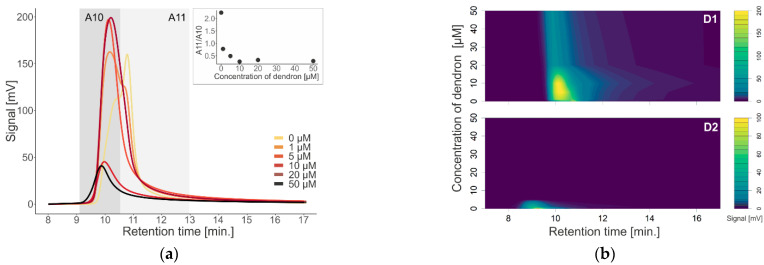
(**a**) Absorption chromatogram of liposomes with different dendron concentrations (0, 1, 5, 10, 20, and 50 µM). Nested graph shows ratio between peak areas at maximum absorption signal at 10 and 11 Minutes. (**b**) Contour plots of liposome absorbance at different dendron concentrations, top—first-generation dendron; bottom—second-generation dendron.

**Figure 3 pharmaceutics-14-01596-f003:**
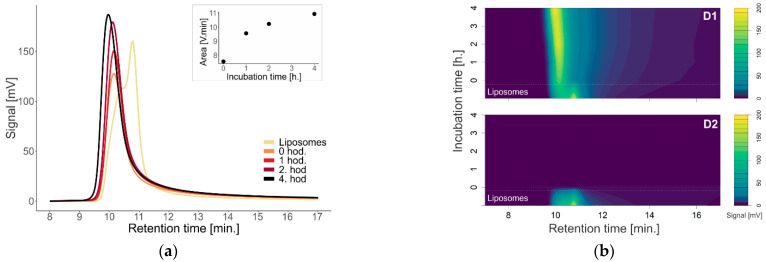
(**a**) Absorption chromatograms of liposomes depending on the time of incubation of added dendron with liposomes at temperature of 37 °C. The nested graph shows the area under the peak of liposomes with dendrons at the corresponding time point. (**b**) Contour plots of liposome absorbance at different incubation times of dendrons with liposomes, top—first-generation dendron; bottom—second-generation dendron.

**Figure 4 pharmaceutics-14-01596-f004:**
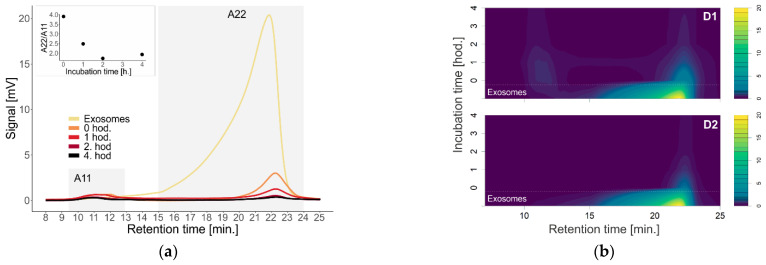
(**a**) Absorption chromatograms of urinary exosomes depending on the time of incubation of added dendrons with exosomes at temperature of 37 °C. The nested graph shows the ratio of absorbance (A) at a retention time of 11 and 22 min as a function of incubation time at temperature of 37 °C. (**b**) Contour plots of exosome absorbance at different incubation times of dendrons with exosomes, top—first-generation dendron; bottom—second-generation dendron.

**Figure 5 pharmaceutics-14-01596-f005:**
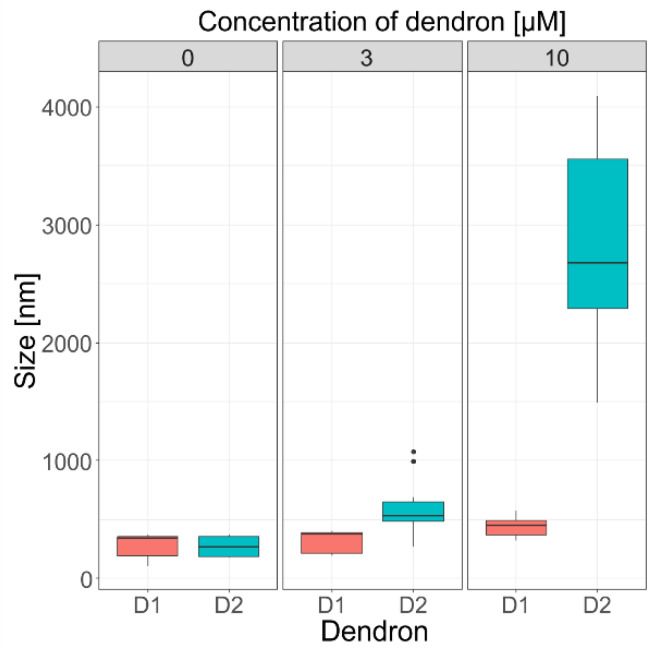
Boxplots showing average Z-size of liposomes with different concentrations of added D1 and D2 dendrons (c = 0 µM, 3 µM, and 10 µM).

**Figure 6 pharmaceutics-14-01596-f006:**
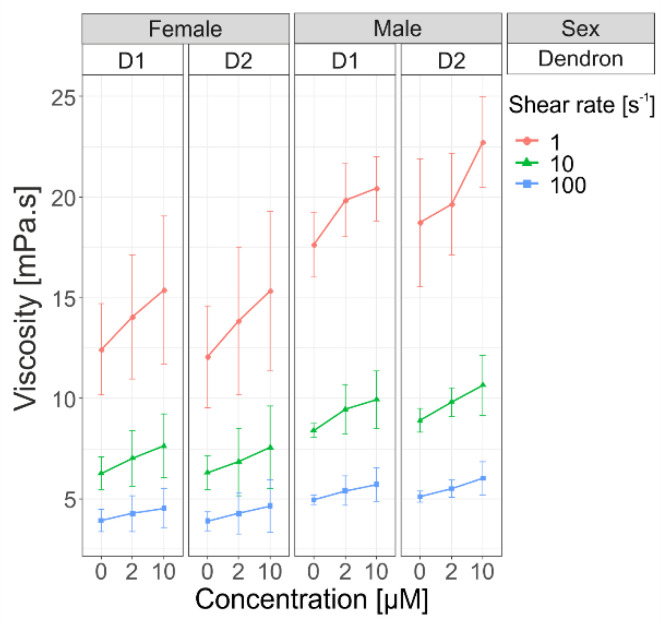
Three−factor ANOVA comparison of viscosities of two amphiphilic phosphorous dendrons (D1 vs. D2). Viscosity was measured at three different concentrations (c_0_ = 0 µM, c_1_ = 2 µM, c_2_ = 10 µM) at three different shear rates (1 s^−1^, 10 s^−1^, and 100 s^−1^) for males (n = 6) and females (n = 6) at 37 °C. The symbols represent the group means at each level of shear rate, while the error bars represent the standard deviation of the underlying data distribution.

**Figure 7 pharmaceutics-14-01596-f007:**
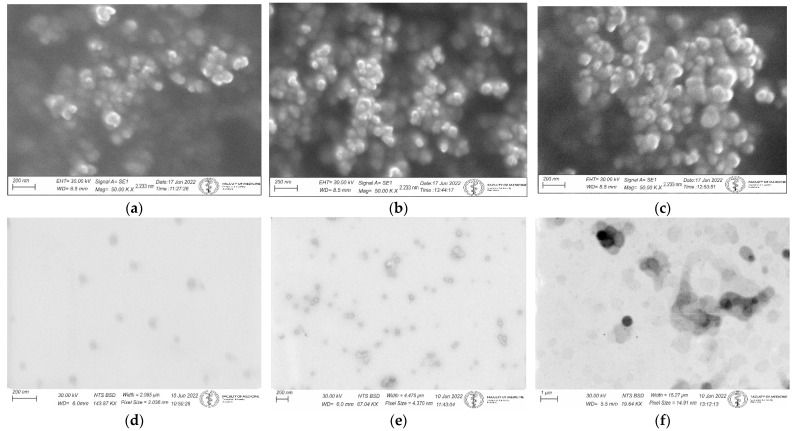
(**a**–**c**) SEM images of secondary electron (SE) of exosomes isolated from blood captured (control) and exosomes with D1 dendrons (concentrations of 2 µM and 10 µM), respectively; (**d**–**f**) SEM images of backscattered electron (BSD) of liposomes captured (control) and liposomes with D1 dendrons (concentrations of 2 µM and 10 µM), respectively.

**Table 1 pharmaceutics-14-01596-t001:** Fluorescence anisotropy of liposomes (control) and after adding 2 µM and 10 µM of D1 and D2 dendrons.

Dendron	Control	D1	D1	D2	D2
Concentration (µM)	-	2	10	2	10
Anisotropy	0.301	0.247	0.215	0.251	0.234

## Data Availability

Data are available upon reasonable request to the corresponding author.

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
