# Peer review of "In Vitro Interactions of Amphiphilic Phosphorous Dendrons with Liposomes and Exosomes—Implications for Blood Viscosity Changes"

_pharmaceutics, 2022, doi:10.3390/pharmaceutics14081596_

Round 1

Author Response

Dear Reviewer,

thank you for careful reading of our text. We appreciate your constructive critique and specific suggestions and we have made a considerable effort to take into account all suggestions proposed by the reviewer when preparing the revised version of the manuscript. Our comments are attached.  

You have also made many salient points that, we hope, helped to improve the narrative of the manuscript. In any case, we are open to consideration of any further comment on our answers.

Reviewer 2 Report

Dear authors, despite the high relevance of the topic and the numerous research methods involved, your work cannot be published without significant revision.

Links to background reviews should be added to the introduction. References must support all statements, except for those derived from the above information. In particular - line 46-50, 66-67, 72-79 and so on.

The introduction should clearly indicate the purpose of the study, as well as its significance for the entire direction of research. It is not clear from your introduction whether the interaction of dendrimers with model membranes was studied in terms of their possibility to cause cell damage, or whether it was planned to modify liposomes with dendrons to improve their delivery properties. If the former, then the introduction contains a lot of unnecessary information (about the properties of liposomes as carriers).

The statement on line 86 is fundamentally wrong!!!

In the materials and methods in paragraph 2.4, it is necessary to add the characterization of preparations of exosomes obtained from urine (transmission electron microscopy or cytofluorometry with tetraspanins), as well as a description of the preparation of exosomes from the blood used in paragraph 2.10. In addition, the method by which the images of Figure 2 df were obtained should be described, since they could not be obtained using scanning microscopy.

Author Response

Dear Reviewer,

we appreciate your constructive critique and suggestions which we have carefully considered when preparing the revised version of the manuscript. 

Our comments are attached.  

Round 2

Reviewer 1 Report

I want to thank the authors for the extensive explanations given at each comment. The rewriting of the introduction furthermore made the purpose of this research clearer and removed a lot of misunderstandings.

Regarding the answers to Q7-8, where the reasoning for the usage of urine exosomes is explained, it would be nice to add 1-2 sentences in the introduction as well to summarize these reasons (e.g., model of plasma membrane, more homogeneous, large quantities possible). That way, future readers also do not have to guess why urine instead of blood was chosen for the chromatographic measurements.

But overall, I was satisfied with the revisions made. 

Author Response

We are very grateful to the Reviewer for this positive feedback and acknowledge the Reviewer's willingness to spend time on the re-submission. We appreciate all suggestions, including extensive explanations given at some comments, which helped to improve the quality of the article. We believe that rewriting of the introduction furthermore made the purpose of this research clearer and removed a lot of misunderstandings.

Q1: Regarding the answers to Q7-8, where the reasoning for the usage of urine exosomes is explained, it would be nice to add 1-2 sentences in the introduction as well to summarize these reasons (e.g., model of plasma membrane, more homogeneous, large quantities possible). That way, future readers also do not have to guess why urine instead of blood was chosen for the chromatographic measurements

R1: We thank the Reviewer for his/her suggestion. In revised version 2 we included the explanation (lines 229 - 237).

“In our study, exosomes isolated from blood and urine were used as a natural model of cell membranes. Because using blood as a source of exosomes was not always possible due to large volume of samples required, especially for chromatographic measurements, we used urine that is an easily available body fluid, even in large volumes. Moreover, exosomes isolated from urine are also more homogeneous than those from blood (exosomes transported in blood have their origin in various organs, thus, the resulting yield would be more heterogeneous). However, for scanning electron microscopy, where only small volumes of sample are required, exosomes isolated from blood were used to make visualization experiments more consistent with blood compatibility studies.”

Reviewer 2 Report

Dear authors, first of all I want to note the significant work carried out on the text of the article. In this form, the text is perceived much better and much becomes clearer. I also want to apologize for the lack of confidence in the methods of electron microscopy used. I was misled since images similar to Figure 7e-f are usually obtained using the negative contrast method of transmission electron microscopy.

I want to clarify my dissatisfaction with the description of the process of formation of exosomes. The only thing that is obtained by invagination of the plasmalemma is an endocytic vesicle. Which goes to the early endosome, the organelle of the cell, which is always present in it! From the early endosome, the late endosome is obtained, in which intralumenal vesicles accumulate. They are formed by invagination of the membrane of the late endosome (multivesicular body). The phrase about double invagination of the plasmolemma can be deciphered as follows: the sheets of the bilayer differ from each other in the composition of lipids, external from internal. with invagination, the leaves change places, that is, the outer becomes the inner. During the formation of an intraluminal vesicle, a repeated change occurs, and thus the sheets of its bilayer are in the same configuration as the sheets of the plasmolemma. This suggests that exosomes can fuse directly with the plasmalemma. But the phrase you used is fundamentally wrong unless explained in detail. And I insist on changing it in accordance with my explanations, and I also recommend that you familiarize yourself with the relevant literature, in addition to the review you mentioned.

I have one question left to work, concerning the characterization of the dendrons under study. I am interested in their size, whether D1 and D2 are different, whether they form compact particles or a network, and how they look under an electron microscope. If they braid the membranes of exosomes and liposomes, disrupt them, promote aggregation, then they should be visible on electron diffraction patterns. If possible, indicate the dendrons with arrows in Figure 7.

I also ask you to clearly indicate in each case the exosomes of urine or blood were used in the experiment. Since you gave protein characterization only for urine exosomes. And the reasons why it was necessary to use exosomes from two sources.

The closing parenthesis is missing on line 82.

On line 104, I suggest using the word "membrane" blood compounds, since you are not discussing proteins and other blood components.

I have doubts about the validity of using reference 5 when describing liposomes (line 108-117), since the article discusses dendrons as delivery vehicles. I kindly ask you to check.

Author Response

We are acknowledging the Reviewer’s willingness to spend time on the resubmission and send her/his comments which we introduce below in full detail together with our replies. We have made a considerable effort to take into account all suggestions proposed by the Reviewer and believe that our reasoning and changes made in revised version 2 sufficiently improved our manuscript.
